# Two-dimensional inorganic molecular crystals

Wei Han[1,4], Pu Huang[2,4], Liang Li[1], Fakun Wang[1], Peng Luo[1], Kailang Liu[1], Xing Zhou[1], Huiqiao Li [1,3], Xiuwen Zhang[2], Yi Cui [3]\* & Tianyou Zhai [1]\*

Two-dimensional molecular crystals, consisting of zero-dimensional molecules, are very appealing due to their novel physical properties. However, they are mostly limited to organic molecules. The synthesis of inorganic version of two-dimensional molecular crystals is still a challenge due to the difficulties in controlling the crystal phase and growth plane. Here, we design a passivator-assisted vapor deposition method for the growth of two-dimensional $Sb_2O_3$ inorganic molecular crystals as thin as monolayer. The passivator can prevent the heterophase nucleation and suppress the growth of low-energy planes, and enable the molecule-by-molecule lateral growth along high-energy planes. Using Raman spectroscopy and in situ transmission electron microscopy, we show that the insulating α-phase of $Sb_2O_3$ flakes can be transformed into semiconducting β-phase under heat and electron-beam irradiation. Our findings can be extended to the controlled growth of other two-dimensional inorganic molecular crystals and open up opportunities for potential molecular electronic devices.

[1] State Key Laboratory of Material Processing and Die & Mould Technology, School of Materials Science and Engineering, Huazhong University of Science and Technology (HUST), 430074 Wuhan, China. [2] Shenzhen Key Laboratory of Flexible Memory Materials and Devices, College of Electronic Science and Technology, Shenzhen University, Nanhai Avenue 3688, 518060 Shenzhen, Guangdong, China. [3] Department of Material Science and Engineering, Stanford University, 94305 Stanford, CA, USA. [4] These authors contributed equally: Wei Han, Pu Huang. \*email: yicui@stanford.edu; zhaity@hust.edu.cn

Molecular clusters, composed of molecules instead of atoms as building blocks, enable the design of functional materials from the bottom-up with unique and customized properties[1,2]. Assembled by molecular clusters, two-dimensional molecular crystals (2DMCs) with in-plane intermolecular van der Waals (vdW) forces have emerged as the promising materials for next-generation electronics because of their quantum tunneling effect, tunable properties by molecular design, dangling-bond-free surface, and molecularly uniform thinness[3]. Recently, 2DMCs have achieved some novel phenomena such as breaking the Landauer limit in a molecular diode[4], Peltier cooling at molecular scale[5], molecular spinterface[6], and humidity-controlled rectification switching[7]. Nevertheless, most 2DMCs currently studied are self-assembled films of complex organic molecules and only limited to growth on hydroxyl-groups terminated substrate, generally with molecular defects and thermal-instability, impacting device performance[8]. Zero-dimensional (0D) inorganic molecules are more stable candidates with simple structures[9], which have applications in diverse fields, such as spin-photovoltaic[10], optoelectronic[11], and field-effect transistors[12]. The library of 0D inorganic molecules includes $C_{60}$, metal-halide perovskites, and metal chalcogenides, and they have been widely explored in the form of quantum dots and thin films[1,11,13]. Unfortunately, two-dimensional inorganic molecular crystals (2DIMCs) have seldom been studied. Due to the miniaturization and integration trends in the electronic industry, the reliable synthesis of large-area and high-quality 2DIMCs is crucial. However, due to the insolubility of inorganic molecules, it is difficult to obtain large-size self-assembled 2DIMCs.

Recently, $C_{60}$-based inorganic nanosheets were obtained via mechanical exfoliation from layered solid, but the yield is limited and the thickness is uncontrollable[14], which hampered further research. By contrast, vapor deposition can produce high-quality 2D inorganic atomic crystals with controllable size and thickness[15–19]. In this process, inorganic sources grow into 2D crystals via atom-by-atom reaction and then layer-by-layer growth along a preferential crystal orientation with the lowest surface energy. Unfortunately, the molecule-by-molecule growth of 2DIMCs however turns to be much less preferential due to the weaker crystalline anisotropy, where the molecules, from all orientations, are bonded via the van der Waals forces. The absence of orientation preference thus results in the difficulties of growing in a 2D way. This growth tendency can be verified by the previous report[20], in which 0D molecules tend to spontaneously assemble to 1D rods instead of 2D flakes or sheets. Therefore, the conventional vapor deposition approach is still very hard for the growth of 2DIMCs. In the growth mechanism of vapor deposition, the vapor-phase precursors are converted to a solid-state crystal via surface nucleation, surface diffusion, and crystal growth[18], and thus the growth plane is determined primarily by the surface nucleation stage. The high-energy planes usually grow preferentially and low-energy planes will be inhibited. Therefore, controlling the crystal plane of nucleation is the key for the synthesis of 2DIMCs.

Herein, we report the synthesis of ultrathin 2D $Sb_2O_3$ molecular crystals with thickness down to monolayer on mica substrates by passivator-assisted vapor deposition (PAVD). The passivator is found to play a key role in controlling the crystal phase and promoting the growth of $Sb_2O_3$ molecular crystals in a preferential orientation. Both in-situ and ex-situ Raman spectroscopy reveal that $Sb_2O_3$ flakes exhibit a heat-induced reversible structural phase transition. Using in-situ TEM, we have observed that $Sb_2O_3$ flakes undergo an $\alpha \rightarrow \beta$ structural transition under electron-beam irradiation. Our findings provide the growth strategy of 2DIMCs and the mechanism of structural transition, and demonstrate their potential applications in phase-change devices.

## Results

**Growth of 2D $Sb_2O_3$ molecular crystals.** Here, 2D $Sb_2O_3$ molecular crystals were successfully prepared on atomically flat mica substrates by $InCl_3$-assisted (Route 1) and Se-assisted (Route 2) PAVD growth process in a three-zone tube furnace, as depicted in the schematic diagram in Fig. 1a. The 2D $Sb_2O_3$ flakes possess a cubic cell ($a = b = c = 11.15$ Å, space group $Fd$-$3m$) composed of spherical $Sb_4O_6$ adamantanoid cages connected by vdW forces[21] in contrast to the well-known 2D materials (Graphene, $MoS_2$)[18,19], which are bonded via the in-plane covalent bonds (Fig. 1b). Within these $Sb_4O_6$ molecular cages, each O atom is connected by two Sb atoms, while each Sb atom is bonded with three O atoms. Figure 1c illustrates the schematic of PAVD growth of 2D $Sb_2O_3$ flakes. In Route 1, hydrophilic $SbCl_3 \cdot xH_2O$ was chosen as the precursor, which undergoes the following reaction to supply the vapor of intermediate $SbOCl$[22].

$$SbCl_3 \cdot xH_2O \xrightarrow[320\,°C]{Ar} SbOCl \uparrow + 2HCl \uparrow + (x-1)H_2O \uparrow . \quad (1)$$

The $SbOCl$ and $H_2O$ react to become $Sb_2O_3$ under the passivation effect of $InCl_3$.

$$2SbOCl + H_2O \xrightarrow[400\,°C]{InCl_3} Sb_2O_3 + 2HCl \uparrow . \quad (2)$$

The passivation effect of $InCl_3$ is demonstrated by a series of experiments by varying the amount of $InCl_3$ in this chemical vapor deposition (CVD) growth process (see Supplementary Fig. 1). Without the passivation effect of $InCl_3$, $Sb_2O_3$ tends to form thick rods. By fine tuning the amount of $InCl_3$, the growth of low-energy crystal planes is suppressed, and the high-energy planes are promoted, leading to the formation of ultrathin $Sb_2O_3$ flakes. The passivation effect of Se in Route 2 and growth mechanism will be discussed in detail afterwards. Figure 1d, e presents the typical optical images of ultrathin triangular $Sb_2O_3$ flakes deposited on mica by Route 1 and Route 2, respectively, with thickness down to monolayer and lateral size of over 20 μm. The lateral size, thickness, and nucleation density of $Sb_2O_3$ flakes can be controlled by the position of mica substrates (Supplementary Fig. 2). The maximum lateral size of a triangular $Sb_2O_3$ flake is 23 μm (Fig. 1f). The representative AFM images of monolayer (1L), bilayer (2L), and trilayer (3L) $Sb_2O_3$ flakes are shown in Fig. 1g–i (more AFM images are shown in Supplementary Fig. 3). It is noted that the height of a monolayer $Sb_2O_3$ flake is about 0.64 nm, consistent with the theoretical thickness of 6.4 Å of single-molecule layer[23]. Furthermore, this PAVD method can be extended to the growth of other molecular crystals, such as $SbI_3$ (Supplementary Fig. 4)[24].

To identify the crystal phase and quality of as-grown $Sb_2O_3$ flakes, Raman spectroscopy was carried out using a 532 nm excitation laser. The primitive cell of bulk $Sb_2O_3$ (space group $Fd$-$3m$, No. 227) consists of two formula units of $Sb_4O_6$ and 20 atoms, and thus there are $20 \times 3$ vibrational modes at the $\Gamma$ point in Brillouin zone (BZ):

$$\Gamma = 2A_g + 2A_u + 2E_u + 2E_g + 3F_{2u} + 5F_{2g} + 5F_{1u} + 3F_{1g},$$

where $2A_g$, $2E_g$, and $5F_{2g}$ modes are Raman active[21]. One of them is a translational $F_{2g}$ mode at lower frequency (81 cm$^{-1}$) and the rest are internal modes within the $Sb_4O_6$ adamantoid cage. As shown in Fig. 1j, $^1F_{2g}$, $^2F_{2g}$, $^1A_g$, and $^2A_g$ modes are observed in both ultrathin and bulk $Sb_2O_3$. The Raman spectra of 2D $Sb_2O_3$ flakes with different thicknesses and bulk $Sb_2O_3$ reveal a layer-dependent blueshift trend of every peak frequency with decreasing thickness. Compared with the Raman spectrum of bulk $Sb_2O_3$ ($^2F_{2g} \sim 188.8$ cm$^{-1}$, $^1A_g \sim 252.9$ cm$^{-1}$), there is a clear trend that both $^2F_{2g}$ peak and $^1A_g$ peak move to the higher wavenumber

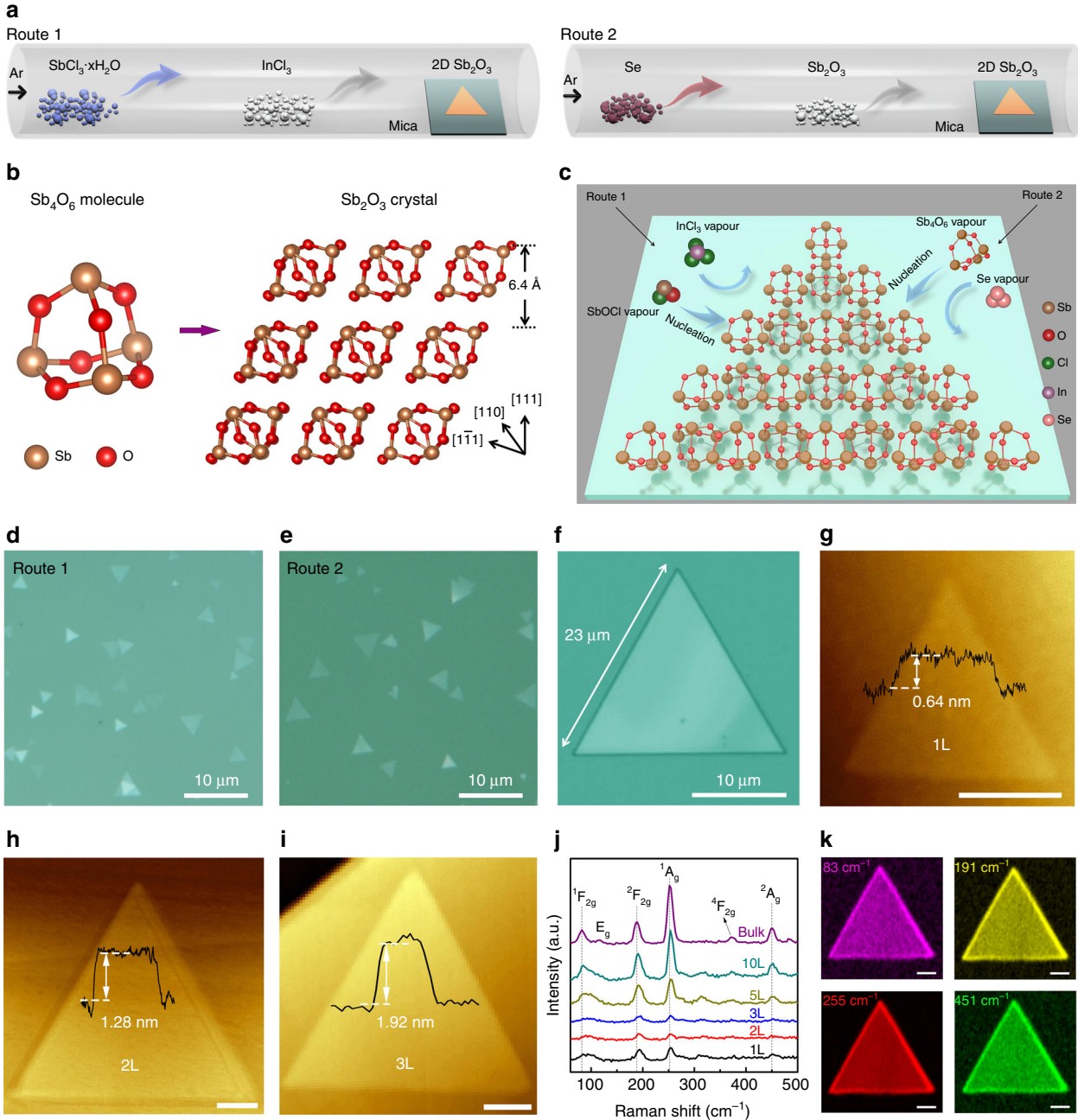

**Fig. 1** PAVD growth of 2D $Sb_2O_3$ molecular crystals. **a** Schematic diagram of passivator-assisted vapor deposition (PAVD) growth of $Sb_2O_3$ flakes in a tube furnace. In Route 1, precursor $SbCl_3 \cdot xH_2O$, undergo a chemical reaction and grow to $Sb_2O_3$ flakes on mica with the assistance of passivator $InCl_3$. In Route 2, precursor $Sb_2O_3$ become gaseous, and grow to $Sb_2O_3$ flakes assisted by passivator Se. **b** Ball-and-stick structural models of spherical $Sb_4O_6$ molecule and $Sb_2O_3$ molecular crystal. **c** Schematic of PAVD growth of 2D $Sb_2O_3$ molecular crystals on mica. The reaction involves: (1) nucleation of $Sb_2O_3$ clusters from decomposition of SbOCl vapor (Route 1) and adsorption of $Sb_4O_6$ vapor (Route 2); (2) adsorption of passivators; (3) horizontal growth of $Sb_2O_3$ flakes along the mica surface. **d**, **e** Optical images of ultrathin triangular $Sb_2O_3$ flakes deposited on mica by Route 1 and Route 2. **f** Optical image of a triangular $Sb_2O_3$ flake as large as ~23 μm. **g–i** Representative AFM images of monolayer (1L), bilayer (2L), and trilayer (3L) $Sb_2O_3$ flakes. Scale bars, 200 nm. **j** Layer-dependent Raman spectra of $Sb_2O_3$ flakes from 1L to bulk. **k** Spatially resolved Raman mapping images of an $Sb_2O_3$ flake at four peak frequencies 83, 191, 255, and 451 $cm^{-1}$. Color scales, black to colorized, low to high Raman intensity. Scale bars, 2 μm

region ($^2F_{2g} \sim 194.0$ $cm^{-1}$, $^1A_g \sim 255.5$ $cm^{-1}$) for monolayer $Sb_2O_3$. Such a blueshift could be a result of the decreased long-range Coulombic intramolecular interactions in thinner flakes[25]. The relation between frequency and number of layers for $^2F_{2g}$ peak and $^1A_g$ peak is plotted in Supplementary Fig. 5a, showing a monotonic decrease trend of frequency with increasing layer number. The molecular vibrations of four Raman modes are illustrated in Supplementary Fig. 5b. The $^1F_{2g}$ mode is an

intermolecular mode and the others are intramolecular ones. The $^1A_g$ mode is from Sb–O–Sb stretch mode while $^2F_{2g}$ and $^2A_g$ are classified into Sb–O–Sb bend modes. The spatially resolved Raman mapping images at four peak frequencies (83, 191, 255, and 451 $cm^{-1}$) reveal the uniform triangular domain of the $Sb_2O_3$ crystal (Fig. 1k).

Figure 2a, b illustrates the top-view ball-and-stick models of monolayer and trilayer $Sb_2O_3$ flakes with (111) plane, which is the

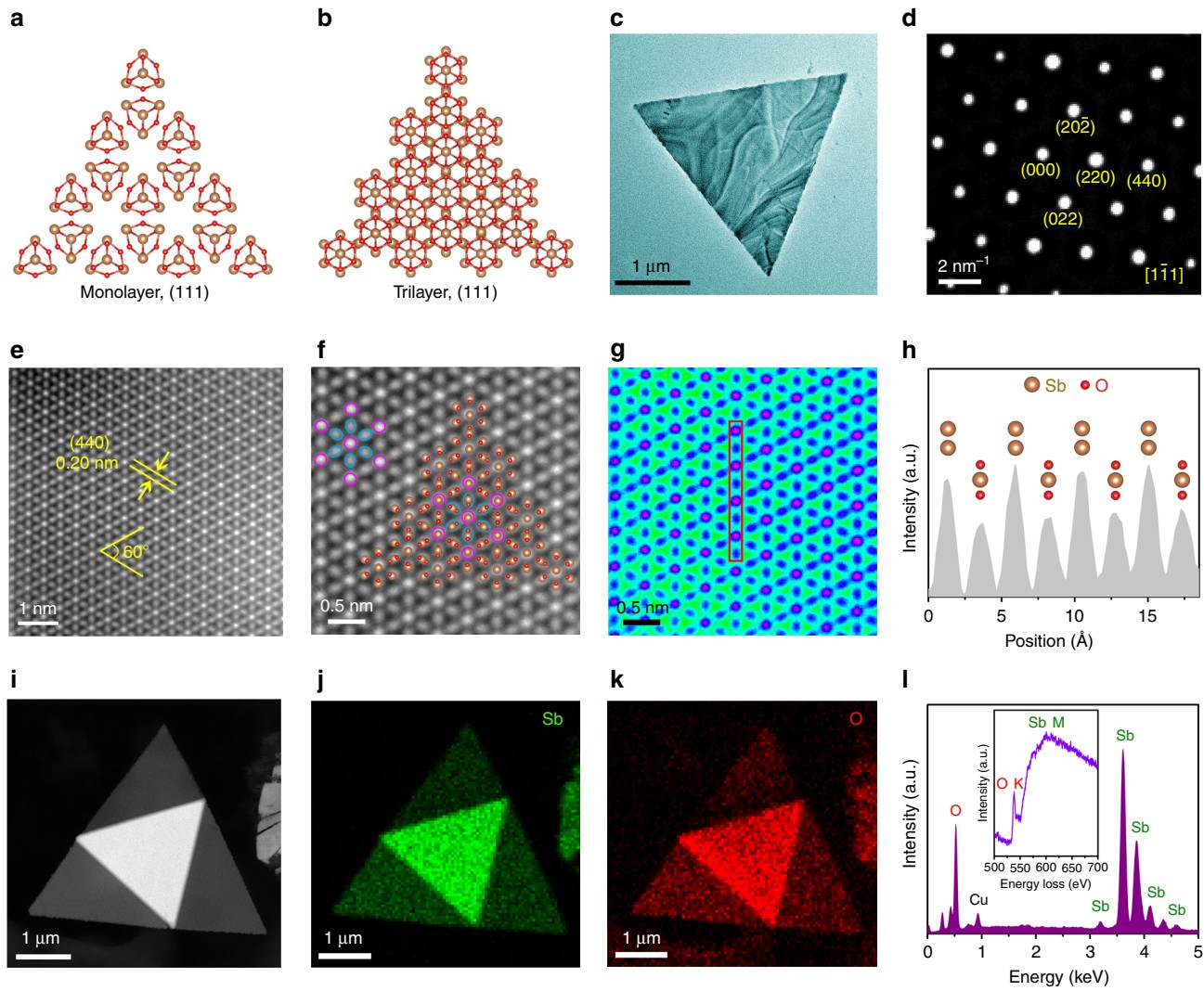

**Fig. 2** Atomic structure of 2D Sb₂O₃ molecular crystals. **a**, **b** Top-view structural models of monolayer and trilayer Sb₂O₃ flakes with (111) plane. Brown balls, Sb atoms. Red balls, O atoms. **c** TEM image of a triangular Sb₂O₃ flake. **d** SAED pattern of the Sb₂O₃ flake. **e** Z-contrast atomic-level HAADF-STEM image of the Sb₂O₃ flake showing the perfect atomic lattice. **f** Enlarged HAADF image and the matched atomic ball model, white atoms are marked by yellow circles and gray atoms are marked by blue circles. **g** Scattered electron intensity color image for (**f**). **h** Intensity line profile along the red box in (**g**). **i**−**k** HAADF image of a stacked flake and the corresponding elemental maps for Sb and O. **l** EDX and EELS spectra of the Sb₂O₃ flake

most close-packed layered crystal plane and has the lowest surface energy for face-centered cubic (FCC) Sb₂O₃. As shown in Fig. 2c, d, the transmission electron microscope (TEM) image and selected area electron diffraction (SAED) pattern reveal that the triangular Sb₂O₃ flake is a high-quality single crystal with $(1\bar{1}1)$ top plane, which belongs to {111} crystal plane family. The specific atomic structure of Sb₂O₃ flake was verified by aberration-corrected high-angle annular dark-field scanning transmission electron microscope (HAADF-STEM). Figure 2e demonstrates the typical Z-contrast atomic-level HAADF image of a ultrathin Sb₂O₃ flake, which reveals a quasi hexagonal close-packed (HCP) structure arrangement of white and gray atoms with crystal plane distance of 0.2 nm corresponding to (440) plane of Sb₂O₃. Figure 2f demonstrates the enlarged HAADF image and the matched atomic ball model, where the brighter round spots marked by red circles represent the Sb atomic columns (red spots in Fig. 2g) and the dimmer elliptic spots marked by blue circles are the staggered O-Sb-O atomic columns (blue spots in Fig. 2g). This lattice pattern can be further verified by the intensity line profile along the red box (Fig. 2h). According to the intensity ratio, the strong peaks represent the two stacked

Sb atoms and the weak peaks are the O-Sb-O atoms. We used energy dispersive X-ray spectroscopy (EDS) and electron energy-loss spectroscopy (EELS) to identify the chemical composition of the flakes. The elemental maps for Sb and O of a stacked flake in Fig. 2i−k show the uniform distribution of Sb and O elements. EDS and EELS spectra in Fig. 2l indicate that the as-grown Sb₂O₃ flakes are composed of Sb and O.

In order to investigate the growth mechanism, the samples obtained by adding different amounts of passivators were characterized. Taking Se as an example (the data about InCl₃ is shown in Supplementary Fig. 1), Sb₂O₃ samples with different morphology were obtained under different amounts of Se. As shown in Fig. 3a−h, the morphology of Sb₂O₃ is transformed from submicronwires (522 nm) to thick triangle flakes (76 nm), and then is changed to thin flakes (9 nm) and ultrathin flakes (less than 4 nm). Confirmed by TEM and Raman characterization (Figs. 3i, j and 2, respectively), the submicronwires and flakes are β- and α-Sb₂O₃, respectively. This indicates that α-Sb₂O₃ tends to grow under the passivation of Se. To explore the passivation mechanism of Se, we calculated the formation energy of β-(001) plane and α-(111) plane with the increase of Se concentration[26].

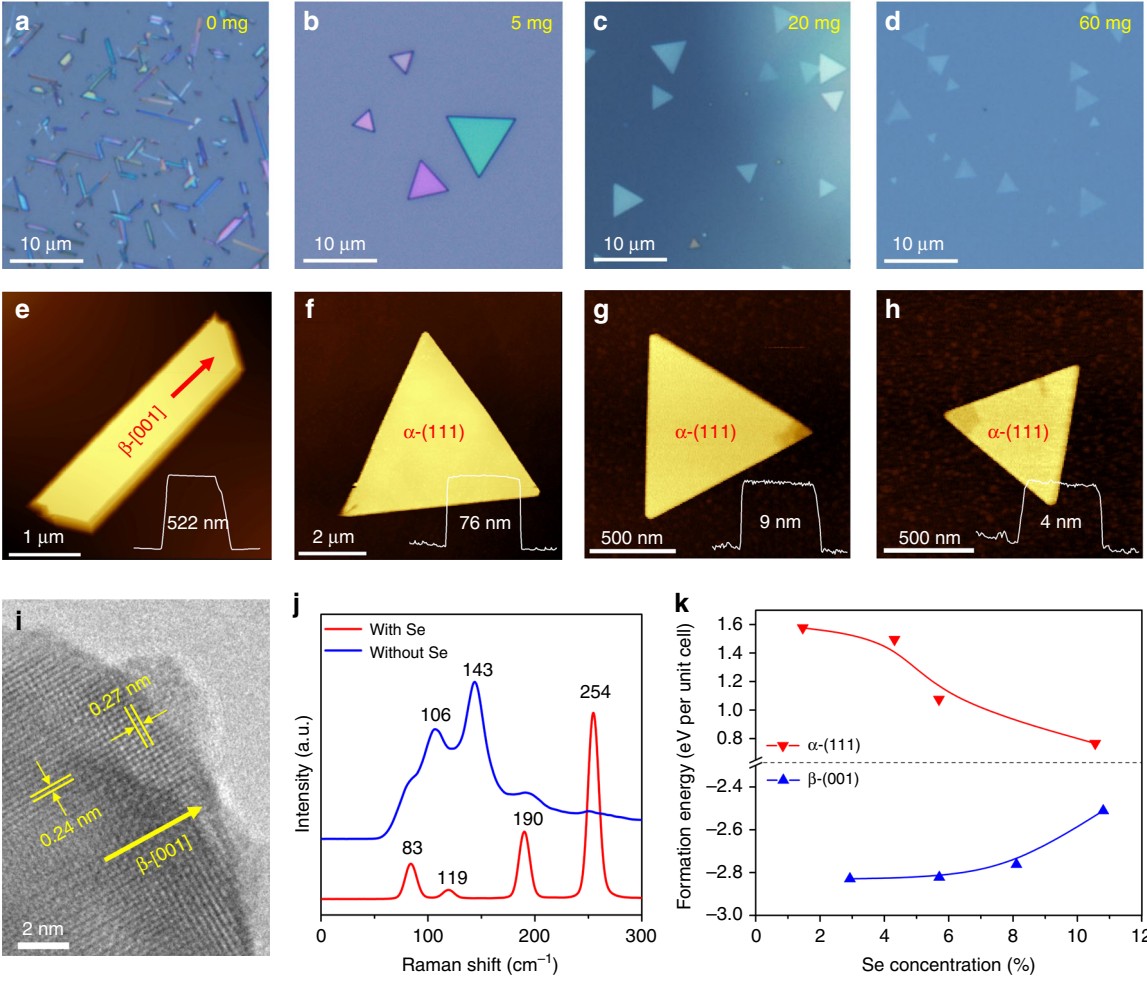

**Fig. 3** Morphology and phase evolution of $Sb_2O_3$ with the amount of Se passivator. **a−h** Typical optical and corresponding AFM images: **a**, **e** 0 mg; **b**, **f** 5 mg; **c**, **g** 20 mg; **d**, **h** 60 mg. The morphology of $Sb_2O_3$ is changed as follows: submicronwires → thick flakes → thin flakes → ultrathin flakes. Phase is changed from β (orthorhombic phase) (**a**) to α (cubic phase) (**b−d**). **i** TEM image of the β-phase $Sb_2O_3$ submicronwire in (**a**) without Se (0 mg); **j** Typical Raman spectra of the submicronwires and flakes. **k** DFT-calculated formation energy of β-(001) plane and α-(111) plane with different concentration of Se passivators, indicating the opposite role played by Se. This relation reveals that the growth of α/β phase is promoted/suppressed with the increase of Se

As shown in Fig. 3k, with the increase of Se concentration, the increased formation energy of β-(001) plane makes the sample unstable, and can hinder (passivate) the 1D growth of $Sb_2O_3$. However, for α-(111) plane, the decreased formation energy can make $Sb_2O_3$ grow favorably on (111) plane with the increase of Se. The above evidence explains why 2D flakes can be obtained under the action of passivator, but does not explain why the thin flakes on (111) plane can be obtained, which is explained by the following calculations.

We compared the surface energies of $Sb_2O_3$ with specific planes (110) and (111) for α phase and (001) for β phase, as is shown in Fig. 4a, b. In the absence of passivators, the (001) plane in β phase of $Sb_2O_3$ exhibits the largest surface energy (255 meV per atom), indicating the high activity and bonding tendency for the surface atoms. Actually, fast growth can obtained for this plane, which leads to the nanorod structural morphology. After using passivators, the gaseous passivator molecules break the dynamic equilibrium. In contrast, the crystal plane (111) of α phase keeps the lower surface energy (38 meV per atom), suggesting the stable surface atomic bonding configuration and slower growth rate. The {110} facets with higher surface energy (73 meV per atom) grow faster (Fig. 4a). As a consequence, the molecular epitaxy mainly occurs along the

{110} planes, resulting in the triangular morphology with (111) top plane for α phase.

As to the situations that $Sb_2O_3$ is passivated with cationic (In) and anionic (Se, Cl) passivators, we find the distinct structure distortions shown in Fig. 4c, d. The anionic Se and Cl passivators induce significant crystal distortions compared with the cationic In[27,28]. Such a phenomenon could be understood by the partial density of states and local charge distribution shown in Fig. 4e. The slight wavefunction overlap between anionic passivators and $Sb_2O_3$ induces strong localized states within the band-gap region, namely shoving up the hybrid orbitals of Se/Cl and $Sb_2O_3$, which will break the pristine thermodynamic equilibrium state and suppress the growth of (111) plane. As to the cationic absorbed situation, valence states present diffused distribution and downward movement overall due to the invasion of In passivator into the cage bonding with O atom tightly, which actually destroys the pristine $Sb_2O_3$ atom stacking configuration[29,30].

**Phase transitions of 2D $Sb_2O_3$ molecular crystals.** It has been reported that the cubic $Sb_2O_3$, denoted as α-$Sb_2O_3$, could be transformed into orthorhombic β-$Sb_2O_3$ (space group *Pccn*, No. 56), which is a reconstructive phase transition from 0D to 1D

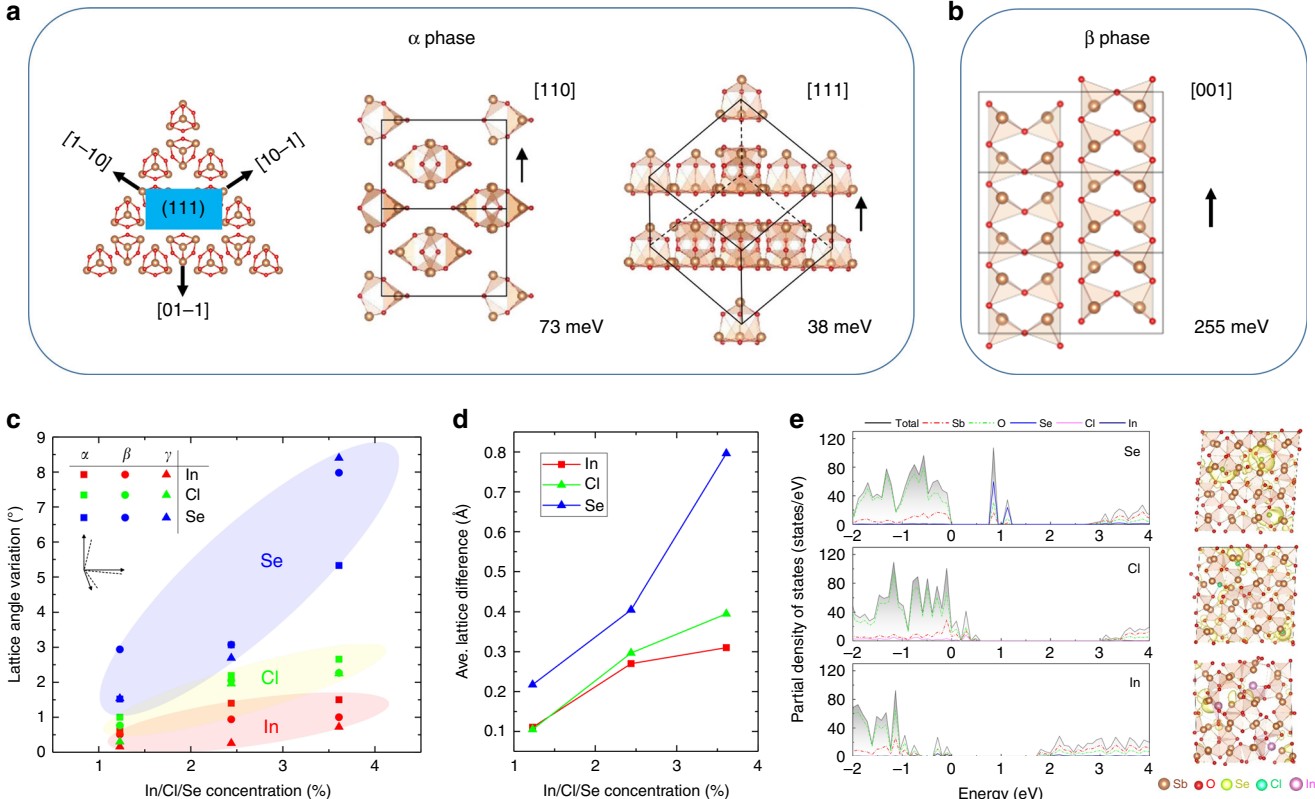

**Fig. 4** DFT-calculation of the effect of passivators. **a** Crystallographic preferential growth orientations and surface energy comparison for specific $Sb_2O_3$ crystal direction (110) and (111) in α phase, and **b** (001) in β phase. Brown balls, Sb atoms. Red balls, O atoms. **c, d** Structural distortions referenced to the pristine $Sb_2O_3$, which is described by the variations of crystal angle (α, β, γ) and average lattice differences ($\Delta_X$, $X =$ Se, Cl and In). **e** Partial density of states for $Sb_2O_3$ with Se, Cl and In absorption on (111) surface and the local charge densities ($\rho = 0.005$ eÅ$^{-3}$). The shaded areas in (**e**) are total density of states

crystal structures[31]. To study the α−β phase transition in $Sb_2O_3$ flakes, we carried out the temperature-dependent in situ and ex situ Raman characterization with the 532 nm laser excitation. Figure 5a shows the in situ temperature-dependent Raman spectra of a thin $Sb_2O_3$ flake during heating process from 293 to 673 K with a step of 20 K. Since $Sb_2O_3$ flakes will sublimate above 673 K, no higher temperature is studied[31]. In the temperature range of 293−433 K, only $^2F_{2g}$ and $^1A_g$ peaks of α-$Sb_2O_3$ are detected. When the temperature rises to 453 K, a new broadening peak appears at 142.7 cm$^{-1}$, which originates from the $A_g$ mode of β-$Sb_2O_3$ confirmed by calculated and measured spectra of orthorhombic phase $Sb_2O_3$[32,33]. In the 453−673 K range, both α and β phases coexist in $Sb_2O_3$ flake, which is a new mixed phase. The α-$Sb_2O_3$ flakes start to sublimate when heated over 673 K, hindering the complete phase transition to β-$Sb_2O_3$. Moreover, $A_g$, $^2F_{2g}$, and $^1A_g$ peaks all exhibit redshift with temperature linearly (Supplementary Fig. 6), which is due to heat-induced enhanced anharmonic phonon−phonon interactions and thermal dilation[34]. Meanwhile, with the increase of temperature, the $A_g$ peak first intensifies and then weakens, and similar phenomenon is observed during the cooling process (Fig. 5b). When the temperature drops to 353 K, only $^2F_{2g}$ and $^1A_g$ peaks are left, indicating that the mixed-phase returns to α-phase. This means $Sb_2O_3$ flake undergoes a partial but reversible phase transition between α and α + β mixed-phase with changing temperature. Figure 5c depicts the corresponding 2D Raman intensity maps plot of temperature converted from Fig. 5a, b. The change of peak position, peak intensity and peak shape can be seen more intuitively from these 2D mapping diagrams. In addition to the change of phonon modes, the relative Raman intensity is usually used to

study the extent and hysteresis of structural transitions. As shown in Supplementary Fig. 7, the relative Raman intensity is defined as the ratio of the Raman intensity of the β-phase to the intensity of entire phases[35]: $R = \beta(A_g)/[\beta(A_g) + \alpha(^1A_g)]$. In the heating process, $R$ increases first and then decreases, and reaches the maximum of 0.66 at 533 K, but decreases to 0.51 at 673 K, indicating an incomplete transition to β phase. When $Sb_2O_3$ flake is cooled down, it remains in the mixed-phase until the temperature drops to 353 K, which results in an interesting thermal hysteresis phase transition. To confirm the mixed phases of α and β occurs in the $Sb_2O_3$ flake, Raman intensity mapping images near 249 cm$^{-1}$ (α-$A_g$) and 141 cm$^{-1}$ (β-$A_g$) were taken at 573 K. As depicted in Fig. 5d, e, the triangle morphology of intensity indicates that the mixed phases are uniform in the flake.

It is reported in the literatures that the complete α to β phase transition of bulk $Sb_2O_3$ appears in the range 829–928 K in different studies[31], and these temperatures are higher than the sublimation temperature of 2D $Sb_2O_3$ flakes (693 K) observed in our study. In order to demonstrate the complete phase transition of $Sb_2O_3$ flakes before their sublimation, we use heat treatment at high temperature for 3 min and then cooling in room temperature to fulfill this transition, because the high temperature phase can be retained during the rapid cooling process (Supplementary Figs. 8−10 and Supplementary Table 1). As shown in Fig. 5f, the ex situ Raman spectra indicate that α-phase can be transformed into β-phase by heat treatment. Below 650 K, there is no obvious phase transition in the sample. When the temperatures of 752 and 807 K are used, the transition from α to mixed phase is observed. At a higher temperature of 823 K, $^2F_{2g}$ and $^1A_g$ peaks disappear completely, with only the peak at 145 cm$^{-1}$ (β-$A_g$) left, which

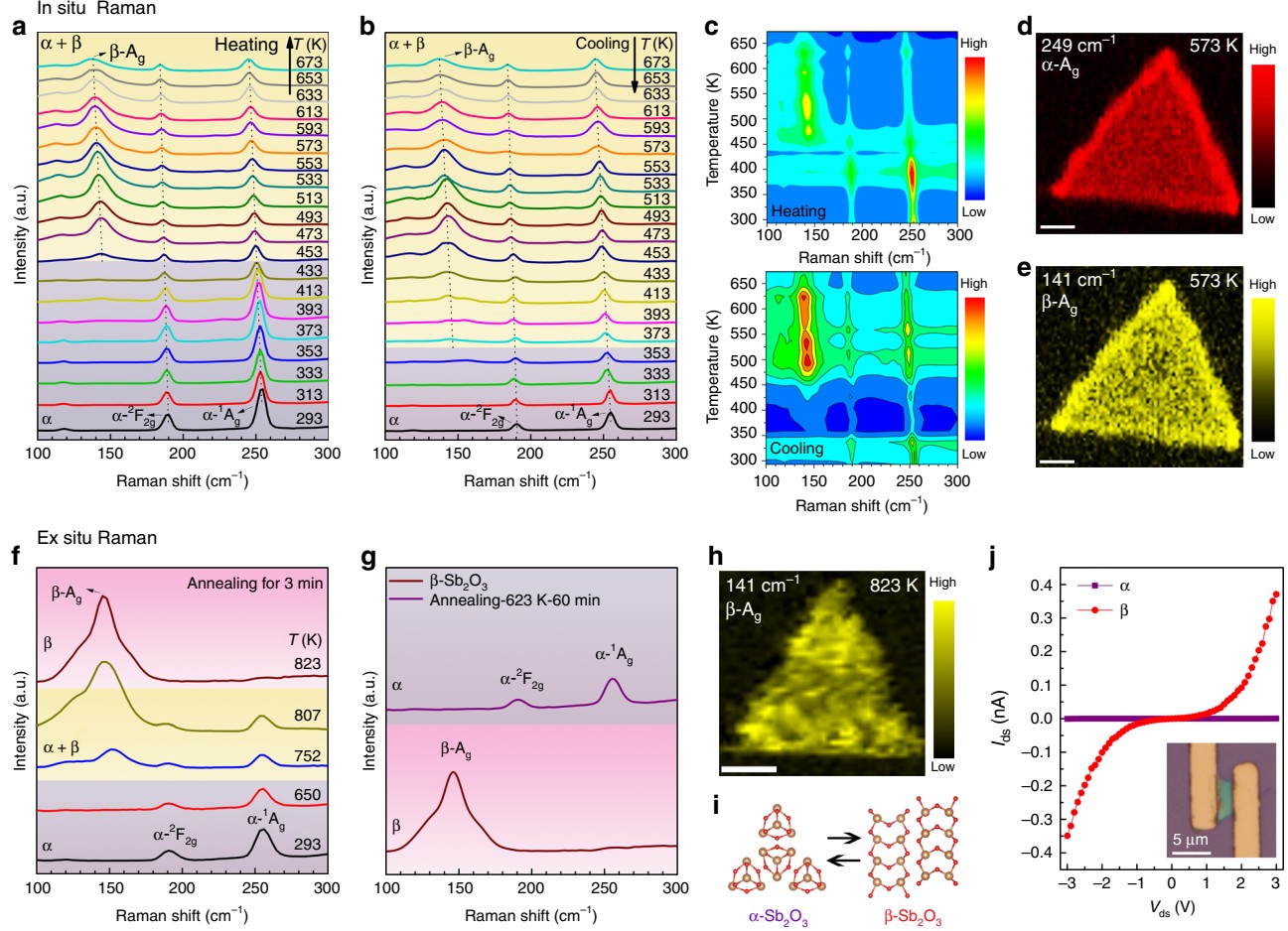

**Fig. 5** Heat-induced phase transition in Sb$_2$O$_3$ flakes. **a, b** In situ temperature-dependent Raman spectra of a thin α-Sb$_2$O$_3$ flake during heating and cooling processes with the temperature window of 293−673 K. **c** The corresponding 2D Raman intensity maps plot of temperature converted from (**a**) and (**b**). **d, e** Raman mapping images near 249 cm$^{-1}$ (α-phase) and 141 cm$^{-1}$ (β-phase) taken at 573 K. Scale bar, 2 μm. **f** Ex situ Raman spectra of an Sb$_2$O$_3$ flake acquired by annealing for 3 min at different temperatures. **g** Raman spectra of the β-Sb$_2$O$_3$ flake before and after annealing for 60 min at 623 K. **h** Raman mapping image near 141 cm$^{-1}$ (β-phase) taken after heat treatment at 823 K. Scale bar, 1 μm. **i** Ball-and-stick structural models of α- and β-Sb$_2$O$_3$. Brown balls, Sb atoms; red balls, O atoms. **j** I–V curves of the Sb$_2$O$_3$ device before (α-phase) and after heat treatment (β-phase). Inset is the optical image of the Sb$_2$O$_3$ device

indicates that the sample has achieved the complete transition from α- to β-phase. This thermal-induced phase transition can be expressed as: α → mixed → β. Figure 5g shows the Raman spectra of the β-Sb$_2$O$_3$ flake before and after heat treatment for 1 h at 623 K, revealing that the reversible transition from β- to α-phase can be realized by re-annealing. The evolution of phases (β → mixed → α) with annealing time is shown in Supplementary Fig. 11. The triangle shape of Raman intensity mapping near 141 cm$^{-1}$ (β-phase) taken after heat treatment at 823 K in Fig. 5h clearly demonstrates that the original morphology remains unchanged after the transition and the phase transition is complete and uniform (Supplementary Fig. 12). Moreover, the flakes after phase transition can be stable in the air for more than 10 months (Supplementary Fig. 13). Figure 5i shows the structural models of α- and β-Sb$_2$O$_3$, which is a transition between 0D Sb$_4$O$_6$ molecules and 1D chains of [SbO$_3$] trigonal pyramids (Supplementary Fig. 14 and Supplementary Table 2). Phase transition is usually accompanied by the change of properties. In order to explore the changes in electrical properties of Sb$_2$O$_3$ flakes, we conducted the I–V measurements of Sb$_2$O$_3$ flake before and after the complete phase transition. The $I_{ds}$−$V_{ds}$ ($I_{ds}$, source-drain current; $V_{ds}$, source-drain voltage) curves of the Sb$_2$O$_3$ flake indicate that the conductivity of the device increases by nearly four

orders (at 3 V) after the complete transition of the flake to β-phase via heat treatment at 823 K (Fig. 5j). This insulator-semiconductor transition from α- to β-phase may be attributed to the decrease of band-gaps (3.0 to 2.1 eV, an indirect-direct band-gap transition, Supplementary Figs. 15, 16)[30]. As discussed above, the anionic (Se, Cl) passivators are critical for the synthesis of 2D Sb$_2$O$_3$ crystals. Although most of these passivators could leave the samples in the form of gas phases of dimers or clusters, in the dilute limit there could be isolated passivator atoms left on the as-grown Sb$_2$O$_3$ flakes. In the molecular α-phase, the appearance of dilute anionic (Se, Cl) passivators will change a small portion of the Sb$_4$O$_6$ cages into cations and keep the crystal insulating. In the nonmolecular β-phase, the anionic (Se, Cl) passivators can be shallow p-type dopants, making the crystal conductive (Supplementary Fig. 17).

**In situ TEM observation of α → β structural transition.** In addition to high temperature, electron-beam irradiation (EBI) is another effective method to induce phase transition[36]. The Sb$_2$O$_3$ flakes were transferred onto Si$_3$N$_4$ windows in a heating chip for in situ TEM (Supplementary Fig. 18). Figure 6a demonstrates the schematic diagram of EBI on the middle of the Sb$_2$O$_3$ flake at room temperature, and the time-dependent in situ TEM images of

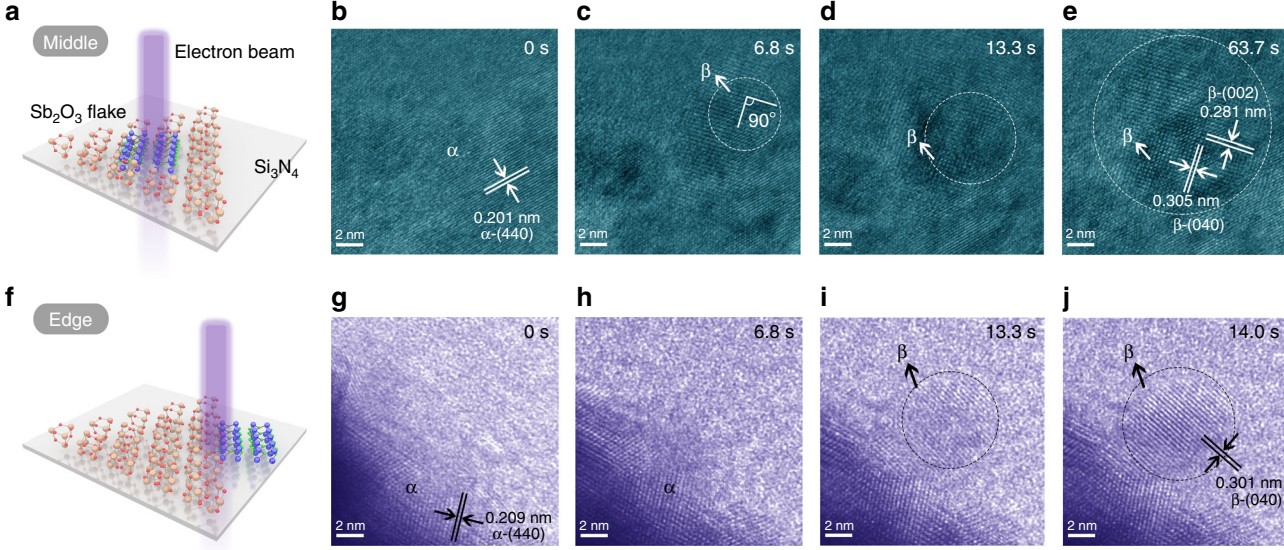

**Fig. 6** In situ TEM study of $Sb_2O_3$ flakes at room temperature. **a** Schematic diagram of electron-beam irradiation on the middle of the $Sb_2O_3$ flake. α-phase: brown balls for Sb atoms, red balls for O atoms. β-phase: blue balls for Sb atoms, green balls for O atoms. **b−e** Time-resolved HRTEM images reveal the β-phase formation at the middle of the $Sb_2O_3$ flake; **f** Schematic diagram of electron-beam irradiation on the edge of the $Sb_2O_3$ flake; **g−j** Time-resolved HRTEM images reveal the β-phase formation at the edge of the $Sb_2O_3$ flake

β-phase formation are shown in Fig. 6b−e and Supplementary Movie 1. At $t = 0$ s, the flake is α-phase (Fig. 6b). After only 6.8 s of EBI, a nucleation of β-phase appears (marked by white circle in Fig. 6c) and this is the starting point of the phase transition. After 13.3 s of EBI, a transition region of β-phase is obtained (Fig. 6d). As the EBI continues, the region of β-phase increases (marked by white circles). As shown in Fig. 6e, the spacing of the two perpendicular planes is 0.281 and 0.305 nm, corresponding to the β-(002) and (040) planes, respectively. The edge of the material is usually different from the middle of the material, here we use EBI to explore the structural phase transition of the edge of $Sb_2O_3$ flake (Fig. 6f). Figure 6g−j demonstrates the TEM images of EBI-induced growth of β-phase nanorod at the edge of the $Sb_2O_3$ flake (Supplementary Movie 2). As a contrast, we compare the images at 6.8 s and find that there is no phase transition at the edge and the phase transition occurs at 13.3 s, which indicates that the phase transition at the edge is slower than that at the middle of the $Sb_2O_3$ flake (Fig. 6h). After a 13.3 s irradiation, a semi-crystalline rod suddenly appears at the edge (Fig. 6i). At $t = 14.0$ s, the semi-crystalline rod is transformed into a single-crystalline rod, and is connected with the edge, indicating that it grows from the flake (Fig. 6j). The crystal plane spacing is 0.301 nm, which is closest to the (040) plane of β-$Sb_2O_3$. This indicates that the flake-nanorod is a heterophase junction fabricated by EBI.

**Atomic mechanism of α → β structural transition in $Sb_2O_3$ flake.** To facilitate phase transition, the flake was heated to 100 −350 °C in TEM, providing thermal activation energy for chemical bond rearrangement and atomic migration. Figure 7a−c presents the HRTEM images of the $Sb_2O_3$ flake at different temperatures. The corresponding atomic models of the α → β phase transition are illustrated in Fig. 7d−f. The initial $Sb_2O_3$ lattice (Fig. 7a, d) is α-phase with (111) top plane and the typical angle of the crystal planes is 60°. At 523 K (Fig. 7b), assisted by the EBI, an intermediate phase (denoted $β_1$) is formed with plane angle of 85° and a phase boundary appears (marked by red dashed line). This phase boundary is also observed in the EBI-induced transition at room temperature (Supplementary Fig. 19). From the arrangement of atomic columns near the red dashed

line, it can be seen that they are not in a straight line, and there are dislocations and defects, which indicates that migration and rearrangement of atoms exist at the boundary. At a higher temperature of 623 K, β-phase is observed and the angle between the two planes is changed to 90° (Fig. 7c, f). This cubic/orthorhombic transition comes from the breakage and recombination of Sb-O-Sb chemical bonds (both α and β phases have six Sb-O bonds per formula unit), which is a typical reconstructive phase transition. From the transformation of plane angles 60° → 85° → 90°, we can speculate a possible mechanism of this α → β reconstructive phase transition which consists of: (i) nucleation of the intermediate phase $β_1$ (Fig. 7b and Supplementary Fig. 20), (ii) migration and rearrangement of Sb and O atoms (Fig. 7e), and (iii) migration of boundaries and growth of β-phase (Fig. 7c). Figure 7e shows the schematic model of migration of atoms near the boundary, and the blue line represents the migrated atomic column, and the blue arrows represent the direction of migration. The migration of Sb atoms from A site to B site (blue letter labeled in Fig. 7e) can increase the angle of the typical crystal planes and lead to the appearance of $β_1$-phase. As a consequence, a β-phase forms, as shown in Fig. 7c. For reconstructive structural phase transition, in addition to angle changes, the spacing and mismatch will also change regularly[37]. Lattice mismatch is given by the expression: $\delta = \frac{|D_\beta - D_\alpha|}{D_\alpha} = \frac{\Delta D}{D_\alpha}$ where $\delta$ is the relative difference of atomic distance ($D_\alpha$ and $D_\beta$) between two adjacent phases. The parameters during the phase transition are listed in Supplementary Table 3. When $\delta$ is between 0.05 and 0.25, a semi-coherent interface is formed ($\delta = 0.084$ in Fig. 7b), which needs dislocations to release the elastic strain energy and achieves the partial matching of atoms in two phases[38].

To conclude, we have prepared ultrathin 2D $Sb_2O_3$ inorganic molecular crystals with thickness down to monolayer by PAVD. The passivator can suppress the heterophase growth, and promote the lateral growth of $Sb_2O_3$ molecular crystals along the high-energy crystal planes. The thermal-induced phase transition of $Sb_2O_3$ flakes is reversible between α- and mixed-phase with a thermal hysteresis loop. The reversible switching between α- and β-phases can be controlled by annealing. Moreover, α-phase can be transformed into β-phase under electron-beam irradiation at

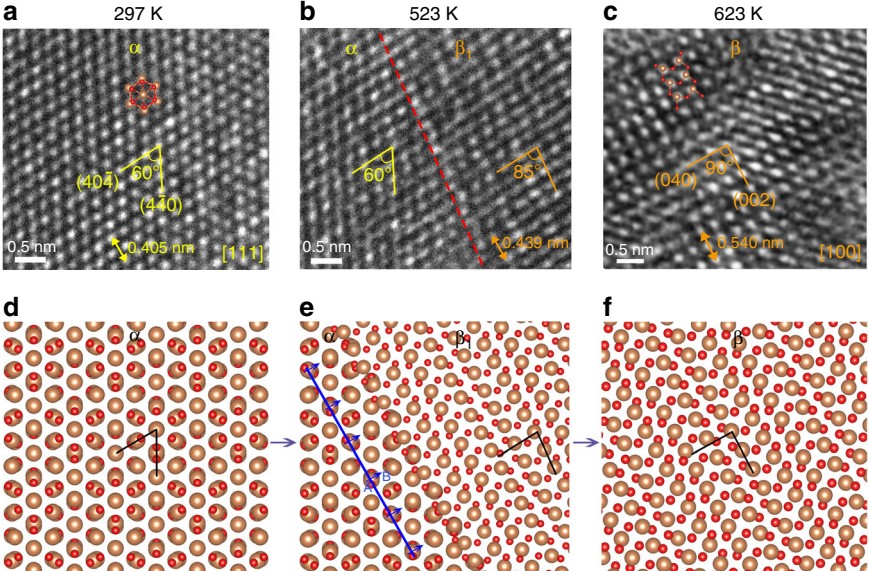

**Fig. 7** Atomic mechanism of structural evolution in $Sb_2O_3$ flake at elevated temperatures. **a** HRTEM image of α-phase with (111) top plane at room temperature. **b** At 523 K, an intermediate phase forms (denoted $β_1$) with plane angle of 85° and a phase boundary appears. **c** At 623 K, β-phase forms with plane angle of 90° and top plane of (100). **d–f** Atomic models of the α → β phase transition corresponding to the TEM images in (**a–c**), respectively

room and high temperature. During this reconstructive phase transition, structural change attributes to the nucleation of the intermediate phase and migration of boundaries. This investigation on 2D $Sb_2O_3$ could be a promoter for the discovery of new 2DIMCs and future phase-change devices.

## Methods

**Growth of 2D $Sb_2O_3$ flakes.** The 2D $Sb_2O_3$ flakes were grown by van der Waals epitaxy methods (Route 1 and Route 2) in a three-zone CVD tube furnace (Thermcraft). In Route 1, 300 mg of $SbCl_3·xH_2O$ (99.9%, Macklin), 15 mg of $InCl_3$ (99.999%, Alfa), and fresh mica ($KMg_3AlSi_3O_{10}F_2$) sheets were put in the zone 1 ($T_1 = 320$ °C), zone 2 ($T_2 = 700$ °C), and zone 3 ($T_3 = 400$ °C), respectively. Ar (60 sccm) was used as carrier gas and the synthesis process was maintained for 45 min. In Route 2, 67 mg of Se (99.999%, Alfa), 40 mg of $Sb_2O_3$ (99.999%, Alfa), and fresh mica sheets were put in the zone 1 ($T_1 = 280$ °C), zone 2 ($T_2 = 650$ °C), and zone 3 ($T_3 = 400$ °C), respectively. Mica substrates were located about 22–26 cm from the center of zone 2. Ar (80 sccm) was used as carrier gas and the synthesis process was maintained for 20 min at ambient pressure. Then the furnace was cooled down to room temperature under the Ar flow.

**Transfer process of 2D $Sb_2O_3$ flakes.** The $Sb_2O_3$ flakes grown on micas were transferred onto TEM Cu grids and $SiO_2$/Si substrates by a water-boiling method. First, the poly (methyl methacrylate) (PMMA) was spin-coated onto the mica substrate with as-grown samples at 3000 rpm for 1 min, and then baked at 150 °C for 5 min. The above process was repeated for three times to obtain a thick robust PMMA membrane on the substrate. Second, the PMMA/mica was put into a 100 ml beaker containing 50 ml of deionized water and heated for 1.5 h at 190 °C. Then, the PMMA membrane with $Sb_2O_3$ flakes was peeled off slowly from the mica substrate by tweezers, and attached to Cu grids or $SiO_2$/Si substrates. Finally, the PMMA/$Sb_2O_3$ was baked at 100 °C for 5 min, and then the PMMA was completely removed in acetone.

**Heat treatment.** We used a home-made slide furnace for rapid annealing heat treatment (Supplementary Fig. 7). Before heating, the α-$Sb_2O_3$ flakes were placed in a quartz tube and outside the heating zone. After the Ar flow washing for an hour, the furnace temperature was increased to the specified temperature at 30 °C/min, then the heating zone was moved to the sample position quickly for rapid heating-up to the set temperature. The actual rapid annealing temperature is defined as the maximum temperature obtained by the sample (Supplementary Fig. 9). After heating for 3 min, the heating zone was moved away from the sample position. The sample was cooled to room temperature in 20 min in Ar flow (heating and cooling rates see Supplementary Table 1). For complete phase transition, the heating rate and cooling rate are 173.3 and 56.2 K/min, respectively. For slow annealing heat treatment, β-$Sb_2O_3$ flakes were slowly heated to 573–623 K at a rate of 20 K/min, then kept for 60 min, and finally cooled to room temperature at a rate of 1 K/min.

**Materials characterizations.** The $Sb_2O_3$ flakes were characterized by OM (Olympus BX51), AFM (Bruker Dimension icon), Raman spectroscopy (WITec Alpha300 Raman and Horiba LabRAM HR800, excitation wavelength of 532 nm), and TEM (FEI Tecnai G2 F30, acceleration voltage of 300 kV). High temperature Raman spectra were collected by ×50 objective lens. Linkam THMS600 and TS1000EV heating stages were used to control the temperature of the sample. The laser power at the top of the stage window is 20–25 mW, correspondingly, the power on the surface of the sample is much smaller. STEM-HAADF images were recorded using Cs-corrected Titan Cubed Themis G2 300 and JEOL JEM-ARM 200 F with acceleration voltage of 200 kV.

**Device fabrication and measurement.** The devices were fabricated by electron-beam lithography (EBL, FEI Quanta 650 SEM, and Raith Elphy Plus) and laser direct-write lithography (LDWL, DMO MicroWriter ML Baby), and then Cr/Au (5 nm/100 nm) metal contacts were deposited using thermal evaporation (Nexdep, Angstrom Engineering). The electrical properties were measured using a semiconductor device analyzer (Keithley 4200-SCS) in a probe station (Lake Shore CRX-6.5K).

**In situ TEM.** The heating holder from DENS solutions was used for in situ heating experiments in a high-resolution TEM (FEI Talos F200X) operated at 200 kV. The $Sb_2O_3$ flakes were transferred onto silicon nitride windows in a heating chip (Wildfire Nanochip) by the PMMA-assisted method. Before in situ TEM experiments, the samples were cleaned with argon plasma for 30 s to remove the residual PMMA and impurities on the surface. The heating rate of the whole process was set to 10 °C/min. During the in situ experiments, the electron-beam current was kept at 50–100 pA.

**DFT calculations.** DFT calculations were performed using the Vienna Ab Initio Simulation Package (VASP)[39,40]. The generalized gradient approximation with the Perdew−Burke−Ernzerhof exchange-correlation functional (GGA-PBE)[41] were used with a 500 eV cut-off energy to expand the electronic wave functions. All the systems with free crystal lattices experienced fully atomic relaxation by the conjugate gradient (CG) algorithm until the residual forces and maximum energy difference were less than $5 \times 10^{-3}$ eV and $10^{-6}$ eV/Å. Van der Waals corrections of optB88-vdW functional[42,43] was considered for dispersion forces, which guarantee the accuracy of structural parameters (the crystal lattice error between theory and experiment is within 0.045‰). The Monkhorst-Pack k-point meshes of $4 \times 4 \times 4$ and $4 \times 4 \times 1$ for $Sb_2O_3$ bulk and 2D flakes with (110) and (111) planes in Brillouin zone were employed. In our study, the passivator molecules were interacted with the surface atoms and molecules of $Sb_2O_3$ randomly with increasing concentrations to simulate the experiment process. All the crystal structures and charge densities were drawn using the VESTA software[44].

## Data availability

All relevant data are either supplied in the paper and Supplementary Information, or available from the authors upon request.

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

## Acknowledgements

We thank J. Luo and W. Xi from Tianjin University of Technology for in situ TEM measurements, data analysis and helpful discussions. This work was supported by the National Natural Science Foundation of China (Grant Nos. 21825103, 51727809, 11774239 and 11804230), Hubei Provincial Natural Science Foundation of China (2019CFA002), the Project funded by China Postdoctoral Science Foundation (Grant No. 2018M642832), and the Fundamental Research Funds for the Central University (Grant No. 2019kfyXMBZ018). The authors also acknowledge the technical support from the Analytical and Testing Center of Huazhong University of Science and Technology. Y.C. acknowledges the support from the Department of Energy, Office of Basic Energy Sciences, Division of Materials Science and Engineering under contract DE-AC02-76SF00515.

## Author contributions

T.Z. and Y.C. conceived the original idea and supervised the research project. W.H. conducted the growth, OM, Raman, AFM, IV, and TEM of 2D crystals. The devices were fabricated by W.H. with L.L. and F.W.'s help. P.H. and X. Zhang carried out DFT calculations and performed the detailed theoretical analysis. P.L. participated in drawing the schematic diagrams. The manuscript was written by W.H., K.L., P.H., X. Zhang, T.Z. and Y.C. with input from the other authors. X. Zhou and H.L. discussed the results and commented on the manuscript.

## Competing interests

The authors declare no competing interests.
