## [Peer Review File · Nature Communications]

Reviewers' comments:

Reviewer #1 (Remarks to the Author):

In the manuscript, "Two-dimensional inorganic molecular crystals", W. Han et al. demonstrate synthesis of ultrathin 2D Sb₂O₃ molecular crystals with thickness down to monolayer on mica by passivator-assisted vapour deposition (PAVD) and report heat-induced reversible structural phase transition. Overall, the manuscript is still premature and lack of clear evidences. Moreover, other researches on 2D oxides and epitaxial growth have been reported elsewhere. The phase transition of Sb₂O₃ might be interesting topic, but I cannot find any novelty or new findings, which can be observed only in this material system. So, I do not think that this manuscript is qualified for publication in Nature Communications.

1. The measured thickness of monolayer Sb₂O₃ is ~ 0.71 nm. It was mentioned that this value is consistent with the theoretical thickness of 6.8 Å of single-molecule layer. This theoretical thickness is calculated from crystal structure or comes from any reference?

2. Other 2D molecular crystals can be produced by this method? Other 2D molecular crystals can be produced by this method? Title of the paper is too broad, even though only Sb₂O₃ was synthesized.

3. In Fig. 1k, why do the edges of Sb₂O₃ have higher Raman intensities at all the Raman shift positions?

4. What is the thickness of the flake shown in Fig. 2c? What makes the winding patterns with different contrast?

5. It is mentioned that the simultaneously evaporated molecules, such as Se, Cl, and In, passivate the surface of Sb₂O₃, leading to lateral growth of triangle flakes, not nanorods. The reason for this is explained in terms of the increased interlayer spacing with concentration of passivation molecules. If so, this is more similar to intercalation? Even though DFT calculation shows possibility of interlayer spacing opening, there is no experimental evidence. If this happens in the experimentally grown samples, the increased interlayer spacing can be observed or measured in AFM and cross-section TEM. Moreover, where is the passivation molecules? Can authors see or measure those molecules by using TEM or other methods?

6. In this work, formation of nanorods are prohibited, meanwhile the triangular shape is preferred. This is explained in terms of increase interlayer spacing in the line of 236-241. However, some words are not appropriate. The "epitaxial growth" in the line 236 is not proper because the triangle flakes also have epitaxially grown multilayer structures. Moreover, meaning of "surface separation effect" is not clear. More critically, I cannot understand why the increased interlayer spacing prohibits vertical growth along [111] direction and leads to horizontal growth. This growth behavior should be related to the surface energy or edge energy, not directly to the interlayer spacing. In addition, explanations for effect of In and mixed In/Cl are ambiguous.

7. As shown in Fig. 4d, the transition does not seem uniform. Moreover, at 493K, there should be the mixed phases of α and β . Show two phases in the mapping images. Is there any damage or modification of shape? Due to phase transition involving structural and volumetric changes, there might be distinct changes in shape and morphology.

8. The sublimation temperature of 2D Sb₂O₃ flakes is 693 K. To obtain high-T phase, the annealed samples were quickly cooled as shown in Fig, 4f. But, annealing temperatures required for complete phase transition is much higher than sublimation temperature. It does not make sense. All the flakes should be evaporated above the sublimation temperature before phase transition.

9. In in-situ experiments, β phase appeared at 453 K, meanwhile phase transition was observed over 673 K (but, no data for 673 K in Fig. 4f) in ex-situ experiments. What make this big difference in phase transition temperatures for two experiments?

10. What are ramping rate and cooling rate for complete phase transition? What is detailed condition of annealing for reverse phase transition in Fig. 4g?

11. To show the change in bandgap after phase transition, optical absorption spectrum or other bandgap measurement should be performed. Electrical measurements in Fig, 4f cannot be a direct evidence for bandgap change and indirect-to-direct band transition. The conductivity of the devices can be influenced by many other factors, not only conductivity of channel material.

Reviewer #2 (Remarks to the Author):

The authors report controlled growth of 2D inorganic molecular crystals (Sb₂O₃) using passivator assisted vapor deposition. This technique is shown to prevent homogeneous nucleation with a fine control of the crystal plane orientation and thickness. They also report a heat induced reversible phase transition from alpha to beta phase under electron-beam irradiation. This general growth strategy of using passivator assisted vapor deposition can lead to discovery of new 2DIMCs that can be explored for novel properties. I find the experimental part quite thorough and interesting and, as the essential component of this manuscript, this may be sufficient to warrant publication in Nature Communications

However, I have some concerns about the theoretical component:

- 1) I find the discussion in the theoretical part (page after fig 3) unclear and confusing. Almost all panels in Fig 3 are obscure and uninformative. What the authors call 'surface binding energy' seems to be a formation energy of a flake _if_ one assumes that the source is a bulk Sb₂O₃ reservoir with chemical potential E_{bulk}/N . Why is this quantity a good descriptor for the present experimental conditions?
- 2) I would expect high-surface energy facets to grow faster, eventually disappearing so that, ultimately, the resultant crystal shape is dominated by the lowest energy crystal planes. If E_{surf} is considered by the authors as a 'surface energy' then I do not understand statements like "Those facets with low surface binding energy (E_{surf}) have a faster growth rate...". This must be clarified and supported by quantitative estimates.
- 3) The authors have many arguments on charge back-transfer and what is energetically favorable (lines 265-275). These arguments have to be supported either by computations or by providing proper references. Moreover, quantifying the lattice distortion is easily accessible using DFT and will give a better insight.
- 4) The authors speculate about the role of passivators in electronic properties of molecular alpha and non-molecular beta phases. These can easily be supported with DFT calculations.

Reviewer #3 (Remarks to the Author):

The synthesis of thin layer antimony oxide is challenging because it can exist in various oxidation state and controlling the growth direction is difficult. Therefore, the synthesis of 2D thin layer Sb₂O₃ has not been studied. However, the authors achieved the formation of 2D Sb₂O₃ layer by using the passivator. It opens up opportunities for the fabrication of the electronic device. The 2D layer Sb₂O₃ is well-characterized overall according to the phase transition. A few points have to be cleared.

1. From the growth of 2D Sb₂O₃ (line 121), hydrochloride gas occurs during the growth. Is there any possibility HCl gas can etch the layer of Sb₂O₃? In AFM images (Supplementary fig.3), the morphology of surface doesn't look clean.
2. If α -Sb₂O₃ can transform to β -phase successfully, the authors would be able to provide the photoluminescence of β -Sb₂O₃ since it can be a semiconductor after annealing.
3. Have the authors checked the stability of the layer Sb₂O₃ after phase transformation? How long

can it last?

Response to the Reviewer #1:

The reviewer's comments: In the manuscript, “Two-dimensional inorganic molecular crystals”, W. Han et al. demonstrate synthesis of ultrathin 2D Sb_2O_3 molecular crystals with thickness down to monolayer on mica by passivator-assisted vapor deposition (PAVD) and report heat-induced reversible structural phase transition. Overall, the manuscript is still premature and lack of clear evidences. Moreover, other researches on 2D oxides and epitaxial growth have been reported elsewhere. The phase transition of Sb_2O_3 might be interesting topic, but I cannot find any novelty or new findings, which can be observed only in this material system. So, I do not think that this manuscript is qualified for publication in Nature Communications.

Authors reply: In order to better understand the growth mechanism, we prepared new samples and carried out more experiments and calculations. On the basis of new experimental evidence, we rebuilt the growth mechanism in our revised manuscript. We believe that the most interesting point about the 2D Sb_2O_3 is its particular 0D structure. The 2D Sb_2O_3 is composed of inorganic molecules, which differs from the 2D oxides with strong in-plane chemical bonds you mentioned. 2D materials composed of atomic 2D sheets (Graphene, MoS_2 , etc.) and of atomic 1D chains (Sb_2S_3 , etc.) have been reported, but the 2D materials composed of 0D inorganic molecules have not been reported to the best of our knowledge. Our finding can be regarded as an accomplishment of the building blocks of 2D materials, from 2D and 1D to 0D. The new structure of 2DIMCs may imply new properties and new applications. As you mentioned “the phase transition of Sb_2O_3 might be interesting topic”, we think our work may open a door for the phase transition study of 2DIMCs, and the phase transition of molecular crystals may be used in phase-change memory device in the future. Moreover, the novelty of our work has been recognized by other reviewers (“quite thorough and interesting”, “opens up opportunities”, and “well-characterized overall”). We hope that our work will be approved by you.

Q1: The measured thickness of monolayer Sb_2O_3 is ~ 0.71 nm. It was mentioned that this value is consistent with the theoretical thickness of 6.8 \AA of single-molecule layer. This theoretical thickness is calculated from crystal structure or comes from any reference?

Authors reply: This theoretical thickness is calculated from the reference (*Acta Cryst.* B31, 2016-2018 (1975)). We have reconfirmed that the theoretical thickness of the monolayer Sb_2O_3 grown on the (111) plane should be 6.4 \AA . We have modified the relevant images and text (Fig. 1b and Line 143). The reference is added as Ref. 23 in the revised manuscript.

Q2: Other 2D molecular crystals can be produced by this method? Title of the paper is too broad, even though only Sb_2O_3 was synthesized.

Authors reply: Other 2D molecular crystals can be also produced by this PAVD method, such as the recently synthesized 2D SbI_3 in our lab. As shown in **Fig. R1**, 2D SbI_3 molecular crystals with flake size over $50 \mu\text{m}$ can be synthesized. We believe that there will be more 2D molecular crystals to be prepared in the future. We have added the relevant images and text in the revised manuscript (Supplementary Fig. 4, and Line 136-137).

Fig. R1 Growth of 2D SbI_3 molecular crystals by PAVD. Structural models of SbI_3 molecule (a) and SbI_3 molecular crystals (b). c, Optical image of a typical hexagonal SbI_3 flake deposited on mica. d, Raman spectrum of SbI_3 flake.

Q3: In Fig. 1k, why do the edges of Sb_2O_3 have higher Raman intensities at all the Raman shift positions?

Authors reply: “Edge-enhanced Raman” is a common phenomenon in 2D materials (*2D Mater.* 2, 035004 (2015)), which may be owing to the higher concentration of oxygen vacancy at the edge than that in the middle. In Fig. 1k, higher density of oxygen vacancies localized at the edges of Sb_2O_3 may lead to the enhanced Raman scattering. Because all the Raman shift positions (83 cm^{-1} , 191 cm^{-1} , 255 cm^{-1} , and 451 cm^{-1}) come from the vibration of Sb_4O_6 molecules (**Supplementary Fig. 5**), so the O vacancies affect the intensity of all peaks.

Q4: What is the thickness of the flake shown in Fig. 2c? What makes the winding patterns with different contrast?

Authors reply: The thickness of the flake in Fig.2c is about 17 nm. The step of heating the sample during the transfer process will lead to the winding patterns, as shown in **Fig. R2e**. And such winding patterns were reported in the previous literatures (*Nano Lett.* 19, 197-202 (2019)).

Fig. R2 Transfer process of Sb_2O_3 flakes for TEM.

Q5: It is mentioned that the simultaneously evaporated molecules, such as Se, Cl, and In, passivate the surface of Sb_2O_3 , leading to lateral growth of triangle flakes, not nanorods. The reason for this is explained in terms of the increased interlayer spacing with concentration of passivation molecules. If so, this is more similar to intercalation? Even though DFT calculation shows possibility of interlayer spacing opening, there is no experimental evidence. If this happens in the experimentally grown samples, the increased interlayer spacing can be observed or measured in AFM and cross-section TEM. Moreover, where is the passivation molecules? Can authors see or measure those molecules by using TEM or other methods?

Authors reply: (1) We have added more experiments and calculations to study the growth mechanism in depth. Based on new evidence, we rebuilt the growth mechanism in our revised manuscript. We think the passivators only interact on the surface of Sb_2O_3 crystals, but not embed in the interlayer, because the interlayer spacing is too small to be embedded. The molecular sizes of InCl_3 and Se are 4.3 Å and 4.7 Å, respectively, which are much larger than the vdW gap of Sb_2O_3 (1.8 Å). So the passivator molecules can't be embedded into the Sb_2O_3 layers. According to the latest experimental evidence, the phase of Sb_2O_3 will change from β to α after using passivators (**Fig. 3**). DFT-calculated formation energy of β -(001) plane and α -(111) plane reveals that with the increase of Se, the growth of β phase is suppressed (increase of formation energy) and the growth of α phase is promoted (reduction of formation energy). We have modified the relevant images and text in the revised manuscript (Fig. 3 and Line 195-222).

(2) If the molecules are embedded into the interlayer, the increased interlayer spacing can be measured in AFM. As shown in **Fig. R3**, after annealing Sb_2O_3 flakes at high Se concentration, the AFM images reveal that the thickness of the Sb_2O_3 flake is unchanged. The cross-section TEM images of Sb_2O_3 flake in **Fig. R4** shows that the spacing is 0.32 nm, corresponding to the (222) plane, and there is no obvious change of interlayer spacing.

(3) The gaseous passivator molecules reacted only on the surface of the Sb_2O_3 crystals with weak interactions, and the passivation molecules exhausted with the carrier gas flow and eventually condensed on the tube outside the high temperature zone. There may be residues of passivator molecules on the surface of the Sb_2O_3 crystals, but the amount is too small to detect by EDS (no In, Cl, and Se signals in Fig. 2l).

Fig. R3 AFM images of a Sb_2O_3 flake before and after reacting at 400 $^\circ\text{C}$ in Se atmosphere for 20 minutes.

Fig. R4 Cross-section TEM images of a Sb_2O_3 flake.

Q6: In this work, formation of nanorods are prohibited, meanwhile the triangular shape is preferred. This is explained in terms of increase interlayer spacing in the line of 236-241. However, some words are not appropriate. The “epitaxial growth” in the line 236 is not proper because the triangle flakes also have epitaxially grown multilayer structures. Moreover, meaning of “surface separation effect” is not clear. More critically, I cannot understand why the increased interlayer spacing prohibits vertical growth along [111] direction and leads to horizontal growth. This growth behavior should be related to the surface energy or edge energy, not directly to the interlayer spacing. In addition, explanations for effect of In and mixed In/Cl are ambiguous.

Authors reply: (1) Based on new experimental data, we rebuilt the growth mechanism in our revised manuscript. The phase of Sb_2O_3 will change from β to α after using passivators (**Fig. 3**). As shown in **Fig. 3g**, with the increase of Se concentration, the increased formation energy of β -(001) plane makes the sample unstable, and can hinder (passivate) the 1D growth of Sb_2O_3 . However, for α -(111) plane, the decreased formation energy can make Sb_2O_3 grow favorably on (111) plane with the increase of Se.

(2) This growth behavior should be related to the surface energy, not directly to the interlayer spacing. So we provide the calculation and discussion about surface energy. As shown in **Fig. 4a,b**, the (001) plane in β phase of Sb_2O_3 exhibits the largest surface energy, indicating the high activity and bonding tendency for the surface atoms. Actually, fast growth can be obtained for this plane, which leads to the nanorod structural morphology. In contrast, the crystal plane (111) of α phase keeps the lower surface energy (38 meV/atom), suggesting the stable surface atomic bonding configuration and slower growth rate. The {110} facets with higher surface energy (73 meV/atom) grow faster (**Fig. 4a**). As a consequence, the molecular epitaxy mainly occurs along the {110} planes, resulting in the triangular morphology with (111) top plane for α phase.

(3) We provide the DFT-calculated structural distortion and electronic characteristics. As shown in **Fig. 4c,d**, we find the distinct structure distortions when Sb_2O_3 is passivated with cationic (In) and anionic (Se, Cl). Moreover, we provide the DFT calculation about the partial density of states for Sb_2O_3 with Se, Cl and In absorption on (111) surface and the local charge densities (See **Fig. 4e**). As shown in **Fig. 4e**, the

passivators have an effect on the density distribution and bonding distribution of Sb_2O_3 . The detailed discussion is added in our revised manuscript.

We have modified the relevant images and text in the revised manuscript (Fig. 3, Fig. 4, and Line 195-256).

Q7: As shown in Fig. 4d, the transition does not seem uniform. Moreover, at 493K, there should be the mixed phases of α and β . Show two phases in the mapping images. Is there any damage or modification of shape? Due to phase transition involving structural and volumetric changes, there might be distinct changes in shape and morphology.

Authors reply: (1) According to your suggestion, we have prepared more new samples. We provide the uniform mapping images of two phases of Sb_2O_3 at 573 K, as shown in **Fig. 5d,e**. (2) The Sb_2O_3 flakes after phase transition maintained the original triangular shape and morphology, and no volumetric changes (**Fig. R5**). But it is possible that the surface will be damaged (sublimation), if the annealing time is too long (over 3 min). However, by controlling the annealing temperature and time, we can obtain the intact β -phase flakes (**Fig. R5a,c**) (only three corners are observed to get slightly sleek). From the height profiles (**Fig. R5b,d**) before and after phase transition, β -phase flakes maintain the original thickness. We have added the relevant images in the revised manuscript (Fig. 5d,e, Supplementary Fig. 12).

Fig. R5 AFM images of α - and β -phase Sb_2O_3 flake (β -phase: annealing at 823 K for 3 min).

Q8: The sublimation temperature of 2D Sb_2O_3 flakes is 693 K. To obtain high-T phase, the annealed samples were quickly cooled as shown in Fig. 4f. But, annealing temperatures required for complete phase transition is much higher than sublimation temperature. It does not make sense. All the flakes should be evaporated above the sublimation temperature before phase transition.

Authors reply: All the flakes should be evaporated above the sublimation temperature, but sublimation takes time. Our annealing time is limited to 3 minutes, and some flakes will not sublimate. In our experiments, the average heating rate and cooling rate for complete phase transition are 173.3 K/min and 56.2 K/min, respectively, much higher than our usual heating (30 K/min) and cooling rates (5 K/min). Therefore, the use of rapid annealing can not only induce the complete phase transition of the sample, but also ensure that the sample does not massively sublimate.

Q9: In in-situ experiments, β phase appeared at 453 K, meanwhile phase transition was observed over 673 K (but, no data for 673 K in Fig. 4f) in ex-situ experiments. What make this big difference in phase transition temperatures for two experiments?

Authors reply: The Raman spectra of Sb_2O_3 flakes are collected at high temperatures in in-situ experiments, while in ex-situ experiments, the spectra are collected at room temperature. Under 673 K annealing condition, the sample has returned to α phase after cooling to room temperature. If Raman spectra can be collected in-situ during rapid annealing, the signal of β phase will be measured, but there is no relevant technical means at present. So, the different test condition makes this big difference in phase transition temperatures.

Q10: What are ramping rate and cooling rate for complete phase transition? What is detailed condition of annealing for reverse phase transition in Fig. 4g?

Authors reply: The ramping rate and cooling rate for complete phase transition are 173.3 and 56.2 K/min, respectively. The detailed condition of annealing for reverse phase transition in Fig. 4g is as follows: β - Sb_2O_3 flakes were heated to 573-623 K at a

rate of 20 K/min, then kept for 60 min, and finally cooled to room temperature at a rate of 1 K/min.

Q11: To show the change in bandgap after phase transition, optical absorption spectrum or other bandgap measurement should be performed. Electrical measurements in Fig. 4f cannot be a direct evidence for bandgap change and indirect-to-direct band transition. The conductivity of the devices can be influenced by many other factors, not only conductivity of channel material.

Authors reply: The PL spectra of α -Sb₂O₃ and β -Sb₂O₃ flakes are shown in Supplementary Fig. 16. α -Sb₂O₃ shows an emission at 525 nm (2.36 eV), which is associated with oxygen vacancy related defect centers in the Sb₂O₃ lattice. The fitted curve shows a strong broadband emission at around 642 nm (~1.93 eV). The PL spectrum indicates that β -Sb₂O₃ is a semiconductor with bandgap about 1.93 eV. The PL intensity of β -Sb₂O₃ is stronger than that of α -Sb₂O₃.

Response to the Reviewer #2:

The reviewer's comments: The authors report controlled growth of 2D inorganic molecular crystals (Sb_2O_3) using passivator assisted vapor deposition. This technique is shown to prevent homogeneous nucleation with a fine control of the crystal plane orientation and thickness. They also report a heat induced reversible phase transition from alpha to beta phase under electron-beam irradiation. This general growth strategy of using passivator assisted vapor deposition can lead to discovery of new 2DIMCs that can be explored for novel properties. I find the experimental part quite thorough and interesting and, as the essential component of this manuscript, this may be sufficient to warrant publication in Nature Communications. However, I have some concerns about the theoretical component:

Q1: I find the discussion in the theoretical part (page after fig 3) unclear and confusing. Almost all panels in Fig 3 are obscure and uninformative. What the authors call 'surface binding energy' seems to be a formation energy of a flake if one assumes that the source is a bulk Sb_2O_3 reservoir with chemical potential E_{bulk}/N . Why is this quantity a good descriptor for the present experimental conditions?

Authors reply: This is because the gaseous passivator molecules react only on the surface of the Sb_2O_3 crystals, and surface energy can give us a better understanding of the passivation mechanism. So, we think surface energy is a good descriptor for the present experimental conditions. Surface energy generally describes the activity of surface atoms. The higher the surface energy, the stronger the atomic activity. In the absence of passivators, the surface energy of β -(001) plane is the highest, and the growth rate is the fastest, so the morphology is wires (**Fig.3**). After using passivators, the gaseous passivator molecules break the dynamic equilibrium, the growth of β phase is suppressed (increase of formation energy) and the growth of α phase is promoted (reduction of formation energy) (**Fig.3g**). The passivators further reduce the activity of (111) plane, and the thin flakes or even monolayer is obtained by suppressing the [111] direction of Sb_2O_3 . According to your suggestion, we have also calculated the formation energy to explain why β phase can only be obtained without passivator and the relationship between the passivator concentration and the Sb_2O_3

thickness, as shown in **Fig.3g**. The detailed discussion is added in our revised manuscript.

Q2: I would expect high-surface energy facets to grow faster, eventually disappearing so that, ultimately, the resultant crystal shape is dominated by the lowest energy crystal planes. If E_{surf} is considered by the authors as a 'surface energy' then I do not understand statements like "Those facets with low surface binding energy (E_{surf}) have a faster growth rate...". This must be clarified and supported by quantitative estimates.

Authors reply: (1) We agree with the reviewer that high-surface energy facets will grow faster, and the resultant crystal shape is dominated by the lowest energy crystal planes. We have modified the corresponding words in our revised manuscript.

(2) We apologize for the mistakes about the statements like "Those facets with low surface binding energy (E_{surf}) have a faster growth rate...". We have modified the corresponding statements to "In contrast, the crystal plane (111) of α phase keeps the lower surface energy (38 meV/atom), suggesting the stable surface atomic bonding configuration and slower growth rate. The {110} facets with higher surface energy (73 meV/atom) grow faster (**Fig. 4a**)".

We have modified the relevant images and text in the revised manuscript (Fig. 3, Fig. 4, and Line 195-256).

Q3: The authors have many arguments on charge back-transfer and what is energetically favorable (lines 265-275). These arguments have to be supported either by computations or by providing proper references. Moreover, quantifying the lattice distortion is easily accessible using DFT and will give a better insight.

Authors reply: We have added relevant computations and references to the revised manuscript. Moreover, according to the reviewer's suggestion, we quantify the lattice distortion by DFT calculation, and we can really get a better understanding.

(1) About the charge back-transfer, we provide the DFT calculation about the partial density of states for Sb_2O_3 with Se, Cl and In absorption on (111) surface and the local charge densities (See **Fig. 4e**). As shown in **Fig. 4e**, The passivators have an effect on the density distribution and bonding distribution of Sb_2O_3 . We also provide the related

references about catalyst-assisted CVD growth (*Adv. Funct. Mater.* 28, 1800181 (2018); *J. Am. Chem. Soc.* 140, 12909–12914 (2018)) in our revised manuscript.

(2) About what is energetically favorable, as we re-verified the experimental results, we corrected the related arguments. On the advice of the reviewer, we have added proper references in our revised manuscript (*J. Phys. Chem. C* 117, 14759–14769 (2013); *J. Solid State Chem.* 213, 116–125 (2014)).

(3) As shown in **Fig. 4c,d**, we quantify the lattice distortions referenced to the pristine Sb_2O_3 , which is described by the variations of crystal angle (α , β , γ) and average lattice differences (Δ_X , X=Se, Cl and In). We find the distinct structure distortions when Sb_2O_3 is passivated with cationic (In) and anionic (Se, Cl). The detailed discussion is added in our revised manuscript.

We have modified the relevant images and text in the revised manuscript (Fig. 3, Fig. 4, and Line 195-256).

Q4: The authors speculate about the role of passivators in electronic properties of molecular alpha and non-molecular beta phases. These can easily be supported with DFT calculations.

Authors reply: (1) For molecular alpha phase, we provide the DFT calculation about the partial density of states for Sb_2O_3 with Se, Cl and In absorption on (111) surface and the local charge densities (See **Fig. 4e**). As shown in **Fig. 4e**, The passivators have an effect on the density distribution and bonding distribution of Sb_2O_3 . The slight wavefunction overlap between anionic passivators and Sb_2O_3 induce strong localized states within the band gap region. (2) For non-molecular beta phase, we give the 2D energy bands of beta phase adsorbed by Cl and Se on the surface (**Fig. R6**). The defect state will appear after adsorption of Se and Cl. The adsorption of Se and Cl introduces p-type defects, which will increase the concentration of p-type carriers and is beneficial to hole conduction. We compared the electronic structure of Alpha phase with that of beta phase, and the change of energy band after lateral adsorption of Se and Cl during the growth of beta phase. Compared with alpha, beta phase has a direct band gap. It can be seen from the k-space band dispersion that the effective mass of the electron hole is relatively small, so it dominates the carrier transport.

Fig. R6 The 2D energy bands of beta phase adsorbed by Cl and Se on the surface. The right is Brillouin.

Response to the Reviewer #3:

The reviewer's comments: The synthesis of thin layer antimony oxide is challenging because it can exist in various oxidation state and controlling the growth direction is difficult. Therefore, the synthesis of 2D thin layer Sb_2O_3 has not been studied. However, the authors achieved the formation of 2D Sb_2O_3 layer by using the passivator. It opens up opportunities for the fabrication of the electronic device. The 2D layer Sb_2O_3 is well-characterized overall according to the phase transition. A few points have to be cleared.

Q1: From the growth of 2D Sb_2O_3 (line 121), hydrochloride gas occurs during the growth. Is there any possibility HCl gas can etch the layer of Sb_2O_3 ? In AFM images (Supplementary fig.3), the morphology of surface doesn't look clean.

Authors reply: HCl gas will not etch the layer of Sb_2O_3 . In our CVD process, the thermal decomposition of $\text{SbCl}_3 \cdot x\text{H}_2\text{O}$ produces HCl gas with about 0.002 mol or 44.8 mL (0.67% in volume) by equation (1), and is less than the reported gas concentration (*ACS Appl. Mater. Interfaces* 7, 15892–15897 (2015)). Moreover, the etching process usually requires plasma gas generated by radio frequency source power at low pressure (*Adv. Mater.* 22, 4014–4019 (2010)), so in our CVD process we think HCl is less likely to etch Sb_2O_3 layers. On your advice, we obtained more AFM images of flakes with different thickness, as shown in **Fig. R7** and Supplementary Fig. 3. The surfaces of the sample and substrate are very clean and smooth, and there are no pits or particles. Furthermore, our TEM data also prove that there is no sign of etching on the surface of the samples (Fig. 2). We have revised Supplementary Fig. 3.

Fig. R7 AFM images of as-grown Sb₂O₃ flakes with clean and smooth surface. (a,c,e) Representative AFM images of Sb₂O₃ flakes with different thickness. Scale bars are 1 μm. (b,d,f) The corresponding height profiles.

Q2: If α - Sb_2O_3 can transform to β -phase successfully, the authors would be able to provide the photoluminescence of β - Sb_2O_3 since it can be a semiconductor after annealing.

Authors reply: Fig. R8 shows the PL spectrum of the β - Sb_2O_3 flake with an excitation wavelength at 532 nm. The fitted curve shows a broad band with a peak wavelength of around 642 nm (~ 1.93 eV). The PL spectrum indicates that the annealed Sb_2O_3 is a semiconductor with bandgap about 1.93 eV, which is less than the calculated bandgap of 2D β - Sb_2O_3 (2.1 eV, Supplementary Fig. 15). The PL spectrum of β - Sb_2O_3 is added in Supplementary Fig. 16.

Fig. R8 Room-temperature PL spectrum of the annealed Sb_2O_3 flake with an excitation wavelength at 532 nm.

Q3: Have the authors checked the stability of the layer Sb_2O_3 after phase transformation? How long can it last?

Authors reply: Yes, we checked the stability of the layer Sb_2O_3 after phase transformation. We measured the Raman spectra of $\beta\text{-Sb}_2\text{O}_3$ flakes after phase transformation on May 27, 2018. After we put the same sample in air for 10 months (15 April, 2019), we tested its Raman spectra again. As shown in **Fig. R9** and Supplementary Fig. 13, the peaks of the $\beta\text{-Sb}_2\text{O}_3$ flakes remain consistent, indicating that the layer $\beta\text{-Sb}_2\text{O}_3$ after phase transformation are air stable, and the stability time is more than 10 months.

Fig. R9 Raman spectra of as-transformed Sb_2O_3 flake (test date: 27 May, 2018) and the same sample after 10 months placed in air (test date: 15 April, 2019).

REVIEWERS' COMMENTS:

Reviewer #1 (Remarks to the Author):

In the revised manuscript, authors fully addressed all the issues raised by the reviewers. Additional experimental and theoretical results improve the manuscript a lot. Now the manuscript is qualified for publication.

Reviewer #3 (Remarks to the Author):

The authors addressed all the question I asked. I believe this manuscript can be published in Nature Communication.

Reviewer #4 (Remarks to the Author):

I was requested to assess the responses to Reviewers #2 regarding the DFT calculations. I find that the authors have answered satisfactorily to all requests and provided extra computational details to prove their point.

Reply to reviewers' comments (NCOMMS-19-05157A)

Reviewer #1 (Remarks to the Author): In the revised manuscript, authors fully addressed all the issues raised by the reviewers. Additional experimental and theoretical results improve the manuscript a lot. Now the manuscript is qualified for publication.

Reply: We sincerely thank the reviewer for offering us with useful suggestions to improve the quality of our work.

Reviewer #3 (Remarks to the Author): The authors addressed all the question I asked. I believe this manuscript can be published in Nature Communication.

Reply: We sincerely thank the comments from the reviewer that helped to greatly improve our manuscript.

Reviewer #4 (Remarks to the Author): I was requested to assess the responses to Reviewers #2 regarding the DFT calculations. I find that the authors have answered satisfactorily to all requests and provided extra computational details to prove their point.

Reply: We really appreciate the reviewer for spending their precious time to assess our manuscript.